# Impact of Bacterial Translocation on Sarcopenia in Patients with Decompensated Cirrhosis

**DOI:** 10.3390/nu11102379

**Published:** 2019-10-05

**Authors:** Cynthia Tsien, Lilia Antonova, Jose Such, Irma Garcia-Martinez, Florence Wong

**Affiliations:** 1Ottawa Hospital Research Institute, Ottawa, ON K1Y 4E9, Canada; lantonova@ohri.ca; 2Department of Medicine, the Ottawa Hospital, University of Ottawa, Ottawa, ON K1H 8L6, Canada; 3Digestive Disease Institute, Cleveland Clinic Abu Dhabi, Abu Dhabi, UAE; such_jos@gva.es; 4Cleveland Clinic Lerner College of Medicine of Case Western Reserve University, Cleveland, OH 44106, USA; 5CIBERehd Hospital general Universitario de Alicante, 03010 Alicante, Spain; irmagarm@gmail.com; 6Instituto de Salud Carlos III, 28029 Madrid, Spain; 7Division of Gastroenterology, Department of Medicine, Toronto General Hospital, University of Toronto, Toronto, ON M5G 2C4, Canada; florence.wong@utoronto.ca

**Keywords:** sarcopenia, bacterial translocation, ascites, cirrhosis, SIRS, hypermetabolism

## Abstract

Advanced liver disease is associated with a persistent inflammatory state, derived from abnormal bacterial translocation from the gut, which may contribute to the development of sarcopenia in cirrhosis. We aim to document the association of chronic inflammation and bacterial translocation with the presence of sarcopenia in cirrhosis. We prospectively followed cirrhotic patients aged 18–70 years with medically refractory ascites at a single tertiary care center in Toronto, Canada. The baseline data included patient demographic variables, the presence of bacterial DNA in serum/ascitic fluid, systemic inflammatory response syndrome (SIRS) status, and nutritional assessment. Thirty-one patients were enrolled, 18 (58.1%) were sarcopenic, 9 (29%) had bacterial DNA in serum and ascites fluid. The mean MELD score was 11.5 ± 4.0 (6–23). Sarcopenic and non-sarcopenic patients did not differ significantly in their baseline MELD scores, caloric intake, resting energy expenditure, the incidence of bacterial translocation, or SIRS. While sarcopenia was not linked to increased hospital admissions or death, it was strongly associated with increased episodes of acute kidney injury (3 vs. 0, *p* = 0.05). This pilot study did not demonstrate an association between sarcopenia and SIRS or bacterial translocation. These results should be confirmed in future larger studies, encompassing a greater number of chronic inflammation events and quantifying levels of bacterial DNA.

## 1. Introduction

Sarcopenia, or loss of muscle mass, is a common complication of advanced liver disease, occurring in 40–68% of cirrhotic patients [1,2]. The presence of sarcopenia in this population has been implicated as a risk factor for serious adverse outcomes of liver disease, including an increased incidence of overt and minimal hepatic encephalopathy [3], longer hospital stays [4], and reduced survival, both pre- [5] and post- liver transplantation [6]. However, the exact mechanisms underlying sarcopenia in this setting are still poorly understood, and likely multifactorial. Some proposed mechanisms have included an observed reduction in caloric intake and/or nutritional absorption [7], altered substrate metabolism [8], and/or inhibition of skeletal muscle protein synthesis [9]. An increasing body of evidence suggests that chronic inflammation may also be an important factor in the development of sarcopenia [10].

It is hypothesized that the persistent inflammatory state observed in advanced liver disease derives from an abnormal bacterial translocation from the gut [11]. When viable bacteria or bacterial products translocate from the intestinal lumen into mesenteric lymph nodes (MLN) and disseminate systemically [12] they produce systemic hemodynamic changes [13] and increased levels of circulating endotoxins, triggering the activation of inflammatory pathways [14]. High levels of inflammatory markers have been found to inversely correlate with muscle mass and strength [15] and to activate a number of molecular pathways related to skeletal muscle wasting [16]. These findings suggest a possible link between bacterial translocation from the gut and the development of sarcopenia in cirrhosis. Previous studies on the contribution of bacterial translocation to liver disease outcomes have pointed to a role in several complications of end-stage liver disease—including hepatic encephalopathy, variceal bleeding, and death [17]. The relationship of gut bacterial translocation to sarcopenia has not been previously studied.

The paucity of data concerning molecular mechanisms of sarcopenia in liver disease, as well as the demonstrated key contribution of sarcopenia to liver disease outcomes, point to an urgent need to research the associated etiological mechanisms. In this pilot study, we aim to document the association of suspected risk factors, including inflammation and bacterial translocation, to the presence of sarcopenia, with the goal of identifying relevant research targets for future investigation.

## 2. Materials and Methods

This study was approved by the Research Ethics Board of the University Health Network, Toronto, Canada (REB #10-0325). Written informed consent was obtained from all patients.

Participants were prospectively recruited at Toronto General Hospital (Toronto, ON, Canada), a single tertiary care center, between September 2010 and March 2015. Eligible participants were aged 18–70 years, diagnosed with decompensated cirrhosis based on liver biopsy or clinical, radiologic, and biochemical findings, and had medically refractory ascites [18] requiring therapeutic paracentesis as part of routine clinical care. We excluded individuals with culture-negative spontaneous bacterial peritonitis (SBP) (ascitic fluid polymorphonuclear (PMN) count ≥ 250/mm^3^), positive ascitic fluid and/or blood cultures, existing renal impairment (creatinine ≥133 umol/L), chronic obstructive pulmonary disease, active malignancy, and individuals who were currently on antibiotics or had been treated with antibiotics within the last two weeks.

Patient data collected included age, gender, and variables related to the etiology and severity of liver disease and liver disease outcomes. The severity of liver disease for each participant was determined by calculating both their Child-Pugh class [19] and MELD score [20].

### 2.1. Bacterial Translocation Measurements

After obtaining informed consent, participants underwent a therapeutic paracentesis under sterile technique, and ascitic fluid (AF) was obtained for cell count, culture (a set of anaerobic and aerobic culture bottles inoculated with 10 mL each of AF), and subsequent analysis for the presence of bacterial DNA (bactDNA), as described below. Peripheral blood cultures were also obtained at the time of paracentesis, as well as blood samples for routine hematologic, biochemical, and coagulation studies.

The ideal location of bactDNA isolation for predicting liver disease outcomes has not been well defined and varies in the literature. Therefore, we assessed bactDNA presence in both blood and ascitic fluid and assessed outcomes related to the DNA findings in either of these sources or in a combination of the two.

DNA isolation, DNA extraction and PCR amplification of the complete 16S ribosomal RNA gene were performed in all serum and AF samples using the previously described methodology [21]. Specimens were processed in airflow chambers and tubes were never exposed to free air. DNA was extracted with a QIAmp DNA Blood Mini Kit (Qiagen, Hilden, Germany) from 200 µL of serum or AF and incubated in a lysozyme-proteinase K buffer for 30 min at 56 °C, and applied onto QIAmp Spin Columns. Samples were microcentrifuged at full speed (13,000 rpm) and DNA was finally eluted with 50 µL of 70 °C preheated AE buffer. The yield and purity of DNA were measured by reading A260 and A260/A280 in a BioPhotometer (Eppendorf). The sensitivity, specificity, and limit of detection of the method have been determined previously [22] with an absolute limit of bactDNA detection of 10 pg/mL.

DNA amplification and PCR reactions for the complete amplification of the 16S ribosomal RNA gene were carried out. Eubacterial primers of a conserved region of the 16SrRNA gene used were: 5′-AGA GTT TGA TCA TGG CTC AG-3′ as forward (located at positions 8–27) and 59-GGT TAC CTT GTT ACG ACT T-39 as reverse (positions 1509–1491). Approximately 10–100 ng of template was added into a reaction mix containing 20 mM Tris HCl (pH 8.4), 50 mM KCl, 1.5 mM MgCl2, 200 mM of each deoxynucleoside triphosphate, 0.4 mM of each primer, and 2.5 U Taq DNA Polymerase (Invitrogen, Life Technologies, Carlsbad, CA, USA) to complete a final volume of 50 µL.

To avoid false-positive results, positive and negative controls were performed in duplicate in each assay. DNA from Escherichia coli was added as a positive control, and sterile water and PCR mixtures (without template) were used as negative controls. PCR was carried out in a Mastercycler personal (Eppendorf) through the cycles as follows: an initial cycle of 95 °C for four minutes was followed by 35 cycles of 94 °C for 30 s, 55 °C for 30 s, and 72 °C for 90 s, with a final extension period at 72 °C for 10 min to complete the cycling sequence. The Total PCR reaction volume was filtered through QIAquick Spin Columns (QIAquick PCR Purification Kit; Qiagen) to remove primers and nucleotides. Purified products were visualized on 1.5% agarose gels stained with ethidium bromide. A band of approximately 1500 base pairs (bp) was obtained, corresponding to the specific amplification of the prokaryotic 16S ribosomal RNA gene.

### 2.2. Inflammation Assessment

Participants were evaluated for systemic inflammatory response syndrome (SIRS) criteria, and had a positive diagnosis if they fulfilled two or more of the following four criteria: (1) Temperature >38 °C or < 36 °C, (2) Heart rate ≥90 beats per minute, (3) Respiratory rate ≥20 breaths per minute or pCO_2_ < 32 mmHg, (4) White blood cell count >12.0 × 10^9^/L or <4.0 × 10^9^/L [23].

### 2.3. Nutritional Assessment

Assessment of the participants’ overall nutritional status was performed using the subjective nutritional assessment (SGA), a scoring system based on clinical and physical exam findings [24]. The SGA questionnaire was used to classify individuals as well-nourished (SGA class A), suspected or moderately malnourished (SGA class B), and severely malnourished (SGA class C). The caloric intake was determined using calorie counts. Assessment of a participant’s body composition using a 2-compartment model of fat mass (FM) and fat-free mass (FFM) was performed using skin-fold measurements, i.e., anthropometrics, and bioelectrical impedance analysis (BIA) (which measures the flow of electric current in different body tissues such as fat and muscle). BIA was performed using a 4-terminal impedance analyzer (model BIA-103; RJL-system, Detroit, MI, USA) with the subject supine, arms and legs extended, on a non-conducting surface. Measurements of resistance and reactance were taken using a single-frequency 800 μA current at 50 kHz on the right side of the subject with electrodes attached at standardized positions. Fat-free mass (FFM) was estimated using the manufacturer’s software based on prediction equations, and FFM was subtracted from the total weight to give a calculated fat mass (FM) [25].

The presence of hypermetabolism, as defined by an elevated corrected resting energy expenditure (REE % predicted), was measured via indirect calorimetry (described in the next section).

In all subjects, body height was measured to the nearest 0.5 cm and body weight was measured to the nearest 0.1 kg after therapeutic paracentesis, but we did not correct for any pedal edema. Body mass index (BMI) was calculated as weight (kg) divided by height (m^2^) (kg/m^2^). A three-day dietary recall was used to assess individuals’ oral intake [26].

Anthropometric measurements were performed by a single investigator (CT) who measured the mid-arm circumference (MAC) and triceps skinfold thickness (TST) to the nearest 1.0 mm at a point midway between the acromion and medial epicondyle of the humerus using a Lang SF caliper (Cambridge Scientific Industries, Cambridge, MA, USA). The MAC was used to calculate the mid-arm muscle circumference (MAMC) using the formula MAC − (TST × 0.314) [27]. Patients with either MAMA or MAMC <5th percentile of age- and gender-matched norms were considered sarcopenic.

### 2.4. Metabolic Assessment

In order to measure resting energy expenditure (mREE), indirect calorimetry was performed on all subjects using the Vmax Encore 29 machine (Carefusion Medical Products, Yorba Linda, CA, USA). After an overnight fast, each subject was placed in a supine position and a transparent canopy was placed over the head in order to measure gas exchange (inspired VO_2_ and expired VCO_2_). Once the patient attained steady-state, i.e., a period of ten minutes during which the variation in VO_2_ and VCO_2_ measurements was ≤10% and variation in respiratory quotient (RQ) ≤5%, the metabolic cart measured the gas exchange over another ten-minute period. The RQ, measuring fuel consumption, and mREE were calculated automatically using the manufacturer’s equations (RQ = VCO_2_/VO_2;_ mREE [kcal/day] = 3.941 × VO_2_ [L/min] + 1.106 × VCO_2_ [L/min]) × 1440) [28]. Hypermetabolism was defined as a corrected mREE (% predicted) higher than 110.

### 2.5. Outcome Variables

In order to assess how representative our population is of other liver disease patient populations, we assessed liver disease outcomes for all patients based on sarcopenia status. Since little is known about the role of bacterial translocation in liver disease, we also assessed outcome variables based on bacterial translocation status. Patients were prospectively followed with routine hematologic and biochemical bloodwork and outpatient follow-up every four months for hospitalization, development of complications, or transjugular intrahepatic shunt (TIPS) insertion, until death or liver transplant, or until they were lost to follow-up. All complications, including the development of SBP, acute kidney injury (AKI), hepatic encephalopathy (HE) or variceal bleeds, as well as hospitalizations were documented, including the reason for admission, in-hospital treatment, and discharge diagnosis. Episodes of AKI were defined as either an absolute increase in serum creatinine of ≥26.4 µmol/L or a 50% increase from baseline [29]. Standard criteria for the diagnosis of SBP (≥250 PMNs in ascitic fluid) [18] and HE (West Haven grade ≥ 2) [30] were used.

### 2.6. Statistical Analysis

Demographic, risk factor and outcome variable distributions were calculated using descriptive statistics. Continuous variables are expressed as means and categorical variables as proportions. Pearson Chi-square tests were performed to assess associations between risk factor variables and sarcopenia. P-values of <0.05 were considered statistically significant. Statistical analyses were performed using JMP^®^ (JMP^®^, Version 14. SAS Institute Inc., Cary, NC, USA, 1989–2019).

## 3. Results

### 3.1. Patient Demographics

A total of 68 patients were screened (Figure 1). Thirty-four consecutive patients who fulfilled all inclusion and exclusion criteria were enrolled. Two subjects were subsequently excluded because they were found to have an infection and one subject was excluded from analysis due to incomplete data. The majority of subjects were men (*n* = 21, 65.6%) with a mean age of 56 ± 9 years. The most common cause of cirrhosis was alcohol (*n* = 17, 53.1%) followed by hepatitis B or C (*n* = 7, 21.9%). Twenty-four subjects were classified as Child-Pugh B (*n* = 24, 77.4%) and seven were classified as Child-Pugh class C (*n* = 7, 22.6%). Two patients (6.3%) had a prior history of hepatic encephalopathy, and four (12.5%) had a history of variceal bleed. Twelve patients (37.5%) were on non-selective beta-blockers. The mean MELD score was 11.5 ± 4.0 (6–23). The mean period of follow-up was 12.7 ± 17.5 months.

### 3.2. Demographic, Nutritional and Metabolic Characteristics of Sarcopenic Patients

Sarcopenia was present in 18 (58.1%) subjects. There was no significant difference in the baseline demographic characteristics of patients who were not sarcopenic vs. those who were (Table 1). As predicted, several body composition markers were significantly different in patients without sarcopenia vs. those with sarcopenia, including, respectively: BMI (30.2 ± 7.1 vs. 24.0 ± 3.7, *p* = 0.059), BIA-measured fat mass (30.6 ± 16.0 vs. 16.9 ± 8.1, *p* = 0.059), triceps skinfold thickness (15.6 ± 5.5 vs. 8.2 ± 2.9, *p* = 0.005), midarm muscle area (45.2 ± 12.5 vs. 27.4 ± 5.4, 0.007) and midarm fat area (22.6 ± 8.5 vs. 9.5 ± 3.6, *p* = 0.008). Conversely, patients in the two groups did not differ in total caloric intake or diet composition, as well as measured or corrected (% predicted) resting energy expenditure (Table 1). Both sarcopenic and non-sarcopenic patients were found to be hypometabolic, with lower measured REE than predicted. There was a trend toward the higher occurrence of Class C SGA scores in the sarcopenic group (44.4 vs. 15.4%, *p* = 0.079, respectively).

### 3.3. Possible Risk Factors and Sarcopenia

In our pilot population, liver disease etiology was not found to significantly contribute to sarcopenia status (*p* = 0.860) (Table 2). Neither MELD nor Child-Pugh liver disease scores were found to be significantly different in sarcopenic vs. non-sarcopenic patients. Among liver function indicators, only serum albumin tended to be lower in sarcopenic patients (35.1 ± 5.1 vs. 31.7 ± 5.0, *p* = 0.083) (Table 2). SIRS was diagnosed in only a small proportion of patients in either patient group (7.7% vs. 11.1%, *p* = 0.677, in non-sarcopenic vs. sarcopenic, respectively).

### 3.4. Bacterial DNA (bactDNA)

BactDNA was detected in only serum in six patients (19.4%), in only ascites in nine patients (29%), while another nine patients had detectable bactDNA in both AF and serum (29%). Bacterial translocation was not significantly different between sarcopenic and non-sarcopenic patients, irrespective of location of bactDNA detected (*p* = 0.211 for serum bactDNA, *p* = 0.739 for ascites, and *p* = 0.857 for both serum and ascites) (Table 2). Baseline demographic, biochemical, and clinical characteristics of patients demonstrating bacterial translocation (bactDNA +ve) and those without bactDNA (bactDNA −ve) were evaluated. Both groups had a similar overall caloric intake (*p* = 0.399), but patients who were bactDNA +ve demonstrated lower protein consumption as compared to patients who were bactDNA −ve (0.042). Indirect calorimetry did not demonstrate a significant difference between groups in measured resting energy expenditure or fuel consumption using RQ. MELD and Child-Pugh scores were not dependent on bacterial translocation status (*p* = 0.389 and *p* = 0.402, respectively), although the presence of bactDNA was associated with lower serum creatinine levels (bactDNA in both serum and ascites: 64.4 ± 8.8 vs. 82.1 ± 21.9, *p* < 0.008, for bactDNA positive vs. negative) and albumin levels (bactDNA in both serum and ascites: 30.4 ± 4.7 vs. 34.2 ± 5.0, *p* = 0.039, for bactDNA positive vs. negative). Despite this, sarcopenia was not more prevalent in patients who were bactDNA positive. SIRS occurred exclusively in patients with bacterial translocation to both ascites and serum.

### 3.5. Clinical Outcomes by Sarcopenia Status

Seven patients in the sarcopenia group underwent transjugular intrahepatic shunt (TIPS) insertion (38.9%) for treatment of refractory ascites, and two patients received liver transplant (11.1%), compared to six (46.2%) and zero patients in the non-sarcopenic group (*p* = 0.981 and *p* = 0.132, respectively) (Table 3). Patients who received a transplant were censored at the time of transplantation. Two deaths occurred in each of the sarcopenic and non-sarcopenic groups (11.1% vs. 15.4%, *p* = 0.429), all due to complications from liver disease.

Sarcopenia was not found to correlate with hospital admissions, or episodes of SBP, HE, and infections (Table 3), but was found to be significantly linked with a higher occurrence of AKI (*p* = 0.05). Time to the event (including transplant, death or loss to follow-up) based on sarcopenia status is presented in Figure 2.

### 3.6. Clinical Outcomes by Bacterial Translocation Status

The presence of bactDNA in blood and ascitic fluid was associated with a significantly higher prevalence of SIRS at baseline as compared to patients who were bactDNA negative (*p* < 0.005). However, during follow-up, this did not translate into significantly increased hospital admissions or episodes of infection (Table 3). During the follow-up period, a total of 11 patients (34.4%) were admitted to hospital, of whom 2 were bactDNA +ve and accounted for 7 hospital admissions, and 9 were bactDNA −ve and accounted for 22 hospital admissions (22.2% vs. 39.1%, *p* = 0.435). No patients required hospitalization after their outpatient paracentesis. There were no significant differences in episodes of AKI, HE or SBP, based on bacterial translocation status.

A significantly higher number of patients with AF bactDNA received TIPS (*p* = 0.019), while serum bactDNA was associated with an increased number of episodes of HE (*p* = 0.059).

During follow-up, one death (11.1%) occurred in the group of patients with bactDNA present in both blood and AF, compared to 3 deaths (13.6%) in other patients (*p* = 0.545). Two of the patients died within 3 months after study enrolment, and two within 6 months, all from complications of their end-stage liver disease. Presence of bactDNA in serum tended towards a significant association with occurrence of death. Time to the event, including transplant, death or loss to follow-up based on bacterial translocation status is presented in Figure 2.

## 4. Discussion

This is the first study to examine the potential associations between sarcopenia, bacterial translocation, and markers of inflammation in patients with decompensated cirrhosis. We had a unique study population of cirrhotic patients with advanced portal hypertension, but low MELD scores. The mean MELD score was only 11.5, and the majority of subjects had a MELD score of less than 15 (*n* = 25, 78.1%). No patients had a MELD score higher than 24. Despite this, we found that sarcopenia was still present in the majority of our study population (58.1%). Similar to previously published studies, we found no association with sarcopenia and Child-Pugh score [5], with sarcopenia being present amongst patients with both Child-Pugh B and C cirrhosis.

While some working groups define sarcopenia as both a loss of muscle mass and function, there is an established body of literature in patients with end-stage liver disease defining sarcopenia solely as low muscle mass [31]. Although we did not have direct CT measurements of psoas or total cross-sectional abdominal muscle mass, we did have several indirect measures of fat mass and fat-free mass, including anthropometrics and bioelectric impedance analysis. Despite these indirect measures, the prevalence of sarcopenia in our cohort was similar to other studies using direct CT morphometrics [5].

Currently, the pathophysiology of sarcopenia in cirrhosis is unknown. In this study, we looked for associations between sarcopenia and possible etiologic mechanisms such as reductions in caloric intake, increased metabolic rate, altered substrate utilization, or bacterial translocation. However, there were no significant differences in these parameters between cirrhotic patients with and without sarcopenia. This may be because these mechanisms are present in all patients with cirrhosis, and not specific to patients with cirrhosis and sarcopenia. For example, all patients with ascites may experience decreased appetite and early satiety with a subsequent reduction in caloric intake. End-stage liver disease may lead to a lack of glycogen stores, altered substrate utilization and increased gluconeogenesis rates [32], in all cirrhotic patients, not just those with sarcopenia.

We did not find that sarcopenia was associated with significantly higher rates of mortality or hospitalizations. There were significantly more episodes of AKI in patients with sarcopenia, but not other complications of liver disease, such as encephalopathy or infections. This may be due to our small pilot study numbers, although we also had a very distinctive group of cirrhotic patients exhibiting significant portal hypertension, but low MELD scores. Just under half of our subjects went on to undergo liver transplantation or TIPS insertion, which also reduced our follow-up time. Furthermore, while patients with sarcopenia had more episodes of AKI than patients without sarcopenia, both groups had preserved kidney function (mean creatinine 78 +/− 21 umol/L) and no history of SBP, both excellent prognostic factors reflective of the patients’ early decompensated state. Despite this, sarcopenia was extremely prevalent, reflecting the fact that sarcopenia is not dependent on the severity of liver dysfunction. Of note, while there was no significant difference in serum creatinine between sarcopenic and non-sarcopenic patients, serum creatinine level is influenced by muscle mass, and may have overestimated the patients’ true glomerular filtration rate in the sarcopenic cohort. Therefore, the sarcopenic patients may have had more significant renal dysfunction than estimated by the serum creatinine, leading to increased episodes of AKI.

Research into the pathophysiological mechanisms underlying sarcopenia in cirrhosis has shown that sarcopenia develops as a result of dysregulation of muscle protein synthesis and protein breakdown [33]. Human data in compensated cirrhotic patients show an increase in skeletal muscle autophagy, as well as a reduction in skeletal muscle protein synthesis [9]. However, it is unclear what drives these changes in skeletal muscle protein synthesis and breakdown. There is published data showing increased levels of myostatin, a negative muscle growth regulator, in the muscle of patients with compensated cirrhosis [9]. Chronic inflammation has been recently proposed as another contributor to the development of sarcopenia. Sarcopenia and loss of muscle strength have been associated with increased serum concentrations of inflammatory markers, including IL-6, CRP, and TNF-alpha [15].

While the exact molecular mechanisms by which inflammation regulates muscle mass are still under investigation, it has been suggested that such proinflammatory mediators upregulate protein breakdown through the activation of FOXO3a and the ubiquitin-proteasome system [34], and downregulate muscle protein synthesis via reduced activation of the mTORC1 signaling pathway [35]. Ongoing inflammation appears to blunt the stimulatory effect of multiple factors on the mTORC1 pathway in muscle, including amino acids [36]. We did not measure circulating levels of IL-6, CRP and TNF-alpha, but we did look for overt manifestations of inflammation through the systemic inflammatory response syndrome (SIRS) criteria. We did not find a significant difference in the occurrence of SIRS in sarcopenic vs. non-sarcopenic patients, however our analysis was limited by the very small number of patients with diagnosed SIRS in the two populations (two and one, respectively). Interestingly, SIRS occurred exclusively in patients showing bacterial translocation in both AF and serum, reinforcing the previously suggested involvement of bacterial translocation in chronic inflammation. The possible contribution of inflammation and bacterial translocation to the development of sarcopenia in liver disease should be further investigated in a population encompassing a greater number of SIRS events.

Zapater et al. hypothesized that bacterial DNA in AF and serum leads to systemic inflammation through activation of the innate immune system, release of nitric oxide and other soluble inflammatory cytokines, and subsequent increase in liver damage [17]. This is consistent with our finding that SIRS occurred only in the presence of bacterial translocation. Similarly, Frances et al. demonstrated a significant correlation between concentrations of bacterial DNA and serum levels of inflammatory markers such as tumor necrosis factor alpha and nitric oxide [22]. Based on these observations, we hypothesized that bacterial translocation may lead to a SIRS state, causing appetite suppression, lower caloric intake, altered substrate utilization, or hypermetabolism, and thus contributing to sarcopenia. However, we found no significant association between sarcopenia and the presence of bactDNA or hypermetabolism. This lack of association may be explained by the fact that, while 29% of our patients had bactDNA in both blood and ascitic fluid (consistent with previously reported rates of bacterial translocation), only three (9.7%) patients had evidence of SIRS.

The low occurrence of SIRS may also account for our excellent clinical outcomes and relatively lower-than-expected mortality rate. Despite 29 hospitalizations that were required by 11 subjects (34.4%), renal function remained well-preserved, and there were only three episodes of acute kidney injury. Only three patients developed episodes of infection, and another three patients developed episodes of hepatic encephalopathy. In addition, there were only four deaths over the follow-up period (12.9%), which is a significantly better prognosis than has been previously reported in the literature [37]. We did not measure the absolute levels of bactDNA in serum and AF, which may account for the discrepancy between the prevalence of bacterial translocation and prevalence of SIRS in our patient population. Whether low levels of bacterial translocation can be adequately controlled through the host response, and therefore may not result in overt systemic inflammation, increased patient morbidity, and mortality, should be a subject of future research.

This is the first paper to examine the relationship between sarcopenia and markers of bacterial translocation and inflammation. In our patient population, we did not find an association between these factors and the development of sarcopenia, although these results should be confirmed in a larger clinical setting. While sarcopenia has been shown to be an independent predictor of complications of cirrhosis [3], our patients’ morbidity and mortality were significantly better than expected. In more advanced states of decompensated cirrhosis, with concomitant SIRS and high MELD scores, the presence of sarcopenia may have a greater impact on outcomes. Future studies should concentrate on the additive effects of sarcopenia and overt inflammation/SIRS on patient morbidity and mortality, in particular the amount of bactDNA and its effects on a patient population with more advanced decompensated cirrhosis. The impact of the pro-inflammatory response on the molecular modulators of protein synthesis and breakdown in the cirrhotic patient population is another important topic for investigation, as it may lead to the identification of possible future therapeutic targets.

## Figures and Tables

**Figure 1 nutrients-11-02379-f001:**
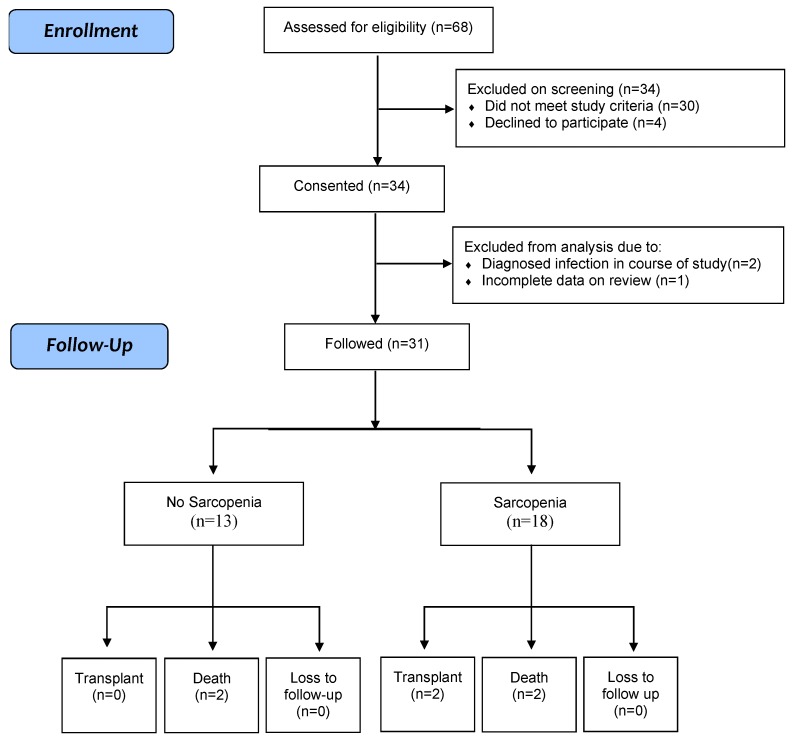
Flow Diagram.

**Figure 2 nutrients-11-02379-f002:**
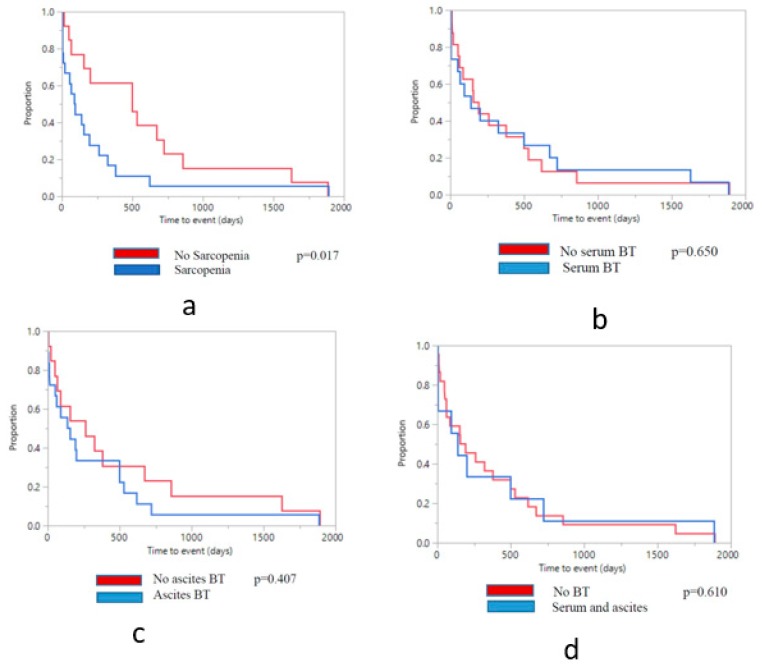
Kaplan-Meier curve for time to the event, including transplant, death or loss to follow-up. (**a**) According to sarcopenia status; (**b**) According to bacterial translocation (BT) status in serum only; (**c**) According to BT status in ascites only; (**d**) According to BT status in both serum and AF.

**Table 1 nutrients-11-02379-t001:** Patient demographic and metabolic characteristics by sarcopenia status.

	Sarcopenia Absent (*n* = 13)	Sarcopenia Present (*n* = 18)	*p*-Value
Age (years) (mean, stdev)	57.9 ± 7.4	55.6 ± 9.7	0.437
Male gender (*n*, %)	8, 61.5	13, 72.2	0.532
BMI (kg/m^2^) (mean, stdev)	30.2 ± 7.1	24.0 ± 3.7	0.059
BIA—Fat mass (kg) (mean, stdev)	30.6 ± 16.0	16.9 ± 8.1	0.059
BIA-Fat-free mass (kg) (mean, stdev)	59.6 ± 18.3	54.8 ± 12.4	0.529
Triceps skinfold thickness (mm) (mean, stdev)	15.6 ± 5.5	8.2 ± 2.9	0.005
Midarm muscle area (cm^2^) (mean, st dev)	45.2 ± 12.5	27.4 ± 5.4	0.007
Midarm fat area (cm^2^) (mean, stdev)	22.6 ± 8.5	9.5 ± 3.6	0.008
Respiratory quotient (mean, stdev)	0.8 ± 0.1	0.8 ± 0.1	0.772
Measured Resting Energy Expenditure (kcal/day) (mean, stdev)	1564.3 ± 345.6	1461.6 ± 258.7	0.584
REE (% predicted) (mean, stdev)	89.8 ± 14.1	93.3 ± 8.7	0.614
Caloric intake (mean, stdev) Carbohydrates (gm) Fat (gm) Protein (gm)	1766.0 ± 633.1 242.7 ± 91.9 53.4 ± 33.1 82.6 ± 39.4	1641.6 ± 458.0 210.3 ± 78.3 54.8 ± 18.2 78.5 ± 31.1	0.920 0.900 0.727 0.993
SGA (*n*, %) Class A Class B Class C	5, 38.5 6, 46.2 2, 15.4	2, 11.1 8, 44.4 8, 44.4	0.072 0.925 0.079

**Table 2 nutrients-11-02379-t002:** Relationship Between Possible Risk Factors and Sarcopenia.

	Sarcopenia Absent (*n* = 13)	Sarcopenia Present (*n* = 18)	*p*-Value
Etiology of liver disease (*n*, %) ETOH Viral NAFLD ETOH/viral Other	7, 53.8 3, 23.1 1, 7.7 2, 15.4 0, 0	9, 50.0 4, 22.2 2, 11.1 3, 16.7 1, 5.6	0.860
Bilirubin (umol/L) (mean, stdev)	31.8 ± 17.9	30.1 ± 26.1	0.712
INR (mean, stdev)	1.30 ± 0.3	1.39 ± 0.3	0.349
Creatinine (umol/L) (mean, stdev)	79.4 ± 22.1	76.2 ± 23.3	0.656
Albumin (mg/dL) (mean, stdev)	35.1 ± 5.1	31.7 ± 5.0	0.083
Sodium (mmol/L) (mean, stdev)	134.9 ± 3.3	134.1 ± 4.2	0.806
MELD score (mean, stdev)	11.4 ± 3.5	11.7 ± 4.5	0.888
Child-Pugh score (mean, stdev)	8.1 ± 1.6	9.0 ± 1.5	0.767
Baseline SIRS (*n*, %)	1, 7.7	2, 11.1	0.677
BT - Ascites (*n*, %)	8, 61.5	10, 55.6	0.739
BT - Serum (*n*, %)	8, 61.5	7, 38.9	0.211
BT -Both Serum and Ascites (*n*, %)	4, 30.8	5, 27.8	0.857

BT = bacterial translocation.

**Table 3 nutrients-11-02379-t003:** Clinical outcomes stratified by sarcopenia and bacterial translocation status.

	Sarcopenia Absent (*n* =13)		Sarcopenia Present (*n* = 18)		*p-Value*	Bacterial Translocation Ascites Only (*n* = 19)		Bacterial Translocation Serum Only (*n* = 15)		Bacterial Translocation Both (*n* = 9)	
	Total Number Events (*n*)	Patients with Outcome (*n*, %)	Total Number Events (*n*)	Patients with Outcome (*n*, %)		Total Number Events (*n*)	Patients with Outcome (*n*, %, *p*-Value)	Total Number Events (*n*)	Patients with Outcome (*n*, %, *p*-Value)	Total Number Events (*n*)	Patients with Outcome (*n*, %, *p*-Value)
AKI	0	0, 0	3	3, 16.7	0.050	2	2, 10.5, 0.765	2	2, 13.3, 0.421	2	2, 22.2, 0.133
Transplants	0	0, 0	2	2, 11.1	0.132	1	1, 5.3, 0.812	1	1, 6.7, 0.962	1	1, 11.1, 0.519
TIPS	6	6, 46.2	7	7, 38.9	0.981	11	11, 57.9,0.019	5	5, 33.3, 0.551	4	4, 44.4, 0.676
Infections	5	2, 15.4	1	1, 5.6	0.364	3	1, 5.3, 0.364	3	1, 6.7, 0.579	3	1, 11.1, 0.865
HE	2	2, 15.4	2	1, 5.6	0.812	3	2, 10.5, 0.812	0	0, 0, 0.059	0	0, 0, 0.232
SBP	3	2, 15.4	1	1, 5.6	0.364	1	1, 5.3, 0.364	1	1, 6.7, 0.579	1	1, 11.1, 0.865
Admission	7	7, 53.8	6	4, 22.2	0.160	15	6, 31.6, 0.531	9	3, 20.0, 0.153	7	2, 22.2, 0.435
Deaths	2	2, 15.4	2	2, 11.1	0.429	1	1, 5.3, 0.846	2	2, 13.3, *0.076*	1	1, 11.1, 0.545

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
