# Peer review of "Impact of Bacterial Translocation on Sarcopenia in Patients with Decompensated Cirrhosis"

_nutrients, 2019, doi:10.3390/nu11102379_

Round 1
Reviewer 1 Report
This is an interesting pilot study and has been well written and described, I have one major comment regarding the methods:
Use of 16S in samples with low microbial abundance has been controversial (e.g. recent disputed findings claiming the existence of microbiomes in the brain or placenta) due to the extremely high risk of either laboratory or reagent contamination (summarised here https://bmcbiol.biomedcentral.com/articles/10.1186/s12915-014-0087-z). I think better description of the methods used for these experiments is required - what controls were used (nuclease free water? the same extraction buffer used for the samples?) Sequencing is mentioned but no data is presented.
Previous studies have applied a similar technique in cirrhosis but are not cited - https://www.sciencedirect.com/science/article/pii/S0016508516355330), from the methods here I'm assuming a simple PCR rather than real-time was run - if so how was presence/absence determined - if using a gel is this sensitive enough? A lot of this could be included in supplementary data but given the controversy in this field I feel better described methods are needed.
Tables 1 and 2 need reformatting as per Table 3 as they are difficult to read currently.
Figure 1 states that for follow-up no events (death or transplant) ocurred in the non-sarcopenic group - this doesn't match with line 210 onwards indicating two deaths in that group (and the K-M in figure 2).There also seems to be a discrepancy between these lines and Figure 1 regarding deaths in the cirrhosis group.
Reviewer 2 Report
The manuscript (number: nutrients-587861), titled: “Impact of bacterial translocation on sarcopenia in patients with 2 decompensated cirrhosis” appears interesting.
The manuscript is well written. However, I have one general comment.
The reference demonstrating that cut-off values of MAMC may identify sarcopenic subjects is lacking.
Authors referred to sarcopenia as a reduced muscle mass, estimated through anthropometry, without considering muscle strength or physical function. The consensus document of the EWGSOP on the definition of sarcopenia set the low muscle mass as the essential criterion for its diagnosis together with low muscle strength or low physical performance (Cruz-Jentoft et al., Age and ageing. 2010;39(4):412-23). Recently, EWGSOP revised the document, suggesting that muscle strength is a criterion for the assessment of sarcopenia, and mass quantity or quality are criteria for confirming this condition in clinical practice (Cruz-Jentoft et al., Age and ageing. 2019;48(1):16-31). Furthermore, International Clinical Practice Guidelines for Sarcopenia (ICFSR) recommend tools suggested by EWGSOP and FNIH (Foundation for National Institutes of Health) for sarcopenia identification (Dent et al., J Nutr health Aging 2018;22(10):1148-1161).
Thus, in my opinion, it is not correct to refer to sarcopenia in this study.
I would recommend to support the presence of the condition of sarcopenia in the population considered in the study by adding data about physical performance or muscle strength. Otherwise, authors should switch to the concept of “low muscle mass” instead of sarcopenia.
Round 2
Reviewer 1 Report
No further comments - my comments on the previous draft have been addressed.
Author Response
Thank you, the reviewer did not have further comments.
Reviewer 2 Report
I do agree that several definitions of sarcopenia are present in the peer-reviewed literature, and we can speculate for a long time whether sarcopenia is a component of physical frailty, or whether they overlap in part or whether physical frailty is a consequence of sarcopenia. Further studies should be performed in order to understand if low muscle mass contributes to low muscle strength and in turn to low physical performance.
I can accept the definition of sarcopenia as the reduction of muscle mass, however it depends on method you use to measure reduction of muscle mass. In the paper you cited (Carey et al. 2019), sarcopenia was defined as low muscle mass according to cut-off values obtained by the gold standard method (CT) and authors concluded that “CT constitutes the best‐studied technique for measuring sarcopenia in patients with cirrhosis”.
In my opinion it would be not correct defined subjects as sarcopenic basing on solely anthropometric measures. Did authors try to identify the two populations (sarcopenic and not sarcopenic) according to SMI estimated by BIA values? There are validated equation and cut-off values (Janssen I, Baumgartner RN, Ross R, Rosenberg IH, Roubenoff R. Skeletal muscle cutpoints associated with elevated physical disability risk in older men and women. Am J Epidemiol. 2004;159(4):413-21).
